# A unified framework for diagnostic test development and evaluation during outbreaks of emerging infections
Madhav Chaturvedi [1,15], Denise Köster [2,15], Patrick M. Bossuyt[3], Oke Gerke [4], Annette Jurke [5], Mirjam E. Kretzschmar [6], Marc Lütgehetmann[7], Rafael Mikolajczyk[8], Johannes B. Reitsma[6], Nicole Schneiderhan-Marra [9], Uwe Siebert[10,11,12,13], Carina Stekly[14], Christoph Ehret[14], Nicole Rübsamen [1,16], André Karch [1,16] ✉ & Antonia Zapf[2,16]

Evaluating diagnostic test accuracy during epidemics is difficult due to an urgent need for test availability, changing disease prevalence and pathogen characteristics, and constantly evolving testing aims and applications. Based on lessons learned during the SARS-CoV-2 pandemic, we introduce a framework for rapid diagnostic test development, evaluation, and validation during outbreaks of emerging infections. The framework is based on the feedback loop between test accuracy evaluation, modelling studies for public health decision-making, and impact of public health interventions. We suggest that building on this feedback loop can help future diagnostic test evaluation platforms better address the requirements of both patient care and public health.

Newly emerging infectious agents present a particular challenge for diagnostic test development and evaluation. These agents often surface in the form of an outbreak, an epidemic or a pandemic with high urgency for targeted infection control, but minimal knowledge about the infectious agents themselves. Rapid availability of diagnostic tests, along with information on their accuracy, however limited, is critical in these situations.

Traditional diagnostic study designs and quality assessment tools developed for individual patient care, as proposed in existing guidelines for diagnostic tests[1–3], are difficult to apply in a volatile environment in which there are continuously evolving research questions, infectious agents, and intervention options. These challenges were particularly apparent during the SARS-CoV-2 pandemic, where the quality of diagnostic studies available in the field was generally limited. The Cochrane review on rapid, point-of-care (POC) antigen and molecular-based tests for diagnosing SARS-CoV-2 infection found a high risk of bias in different domains in 66 of the 78 studies considered (85%)[4]. The most frequent potential source of bias was identified in the reference standard domain, including potential of imperfect gold/reference standard bias, incorporation bias, and diagnostic review bias (an explanation of these biases is given in Table 1). In the Cochrane review on antibody tests for identification of current and past infection with SARS-CoV-2[5], the most frequent potential source of bias was identified in the patient selection domain, due to selection or spectrum bias (48 of 54 studies, 89%). A particular issue is that biases with regard to the reference standard or the index test can lead to an overestimation or underestimation of sensitivity and specificity.

In both application fields, the differences between the diagnostic test accuracy estimates reported by the manufacturers and those estimated later in the Cochrane meta-analyses were enormous. The mean sensitivity reported by manufacturers for antigen tests was 89% (as of 22/06/2022)[6]. In comparison, the sensitivity estimated in the meta-analysis on antigen tests[4] was 72% in symptomatic and 58% in asymptomatic individuals. This discrepancy shows that the timely evaluation of newly developed laboratory tests under real-life conditions is crucial and should be planned and started before market launch.

This Perspective article is the result of an interdisciplinary workshop which we conducted as part of a research project funded by the German Research Foundation. This brought together expertise from all disciplines relevant to diagnostic test development and evaluation, ranging from molecular test development to public-health decision-making. Firstly, the project members gave presentations on the respective sub-areas of the project to create a common basis for the following moderated panel discussions, integrating the expertise and experience from the individual workshop participants. Subsequently, a previously created draft of the framework was further developed and the next steps were planned. We describe the challenges and potential solutions that were discussed for implementing state-of–the-art diagnostic test development and evaluation processes, based on accuracy studies performed at different phases of an epidemic or pandemic.

This Perspective is divided into three key sections. First, we discuss the relevance of diagnostic studies for public health decision-making based on mathematical models. Second, we describe the challenges in developing

**Table 1 | Summary of types of bias typically present in diagnostic accuracy studies**

| Domain | Type of bias | Definition of domain | Description | Example based on the SARS-CoV-2 antigen test |
|---|---|---|---|---|
| Reference standard | Imperfect gold/reference standard bias | "This is the test used to define the target condition, and the underlying assumption is that it reflects the truth. By design, the reference standard is assumed to be flawless."[85] | The reference standard does not correctly classify the target condition. | A single RT-PCR is used as a reference standard. |
| | Incorporation bias | For the evaluation of the SARS-CoV-2 antigen test, the detection of viral RNA via reverse transcription–polymerase chain reaction (RT-PCR) is in general the reference standard. This might be problematic if the target condition is e.g. being infectious. | The definition of the reference standard incorporates results from the index tests. | The true infection status is defined based on both the RT-PCR and the antigen test result. |
| | Diagnostic review bias | | The results of the reference standard are interpreted with knowledge of the results of the index test. | The results of the antigen test are known when the RT-PCR result is interpreted. |
| | Differential verification bias | | Different reference standards are used between groups of patients to determine the true disease state. | Positive antigen test results are verified by RT-PCR and negative results are verified by a second antigen test. |
| | Partial verification bias | | The true disease state is verified by a reference standard only for a subgroup. | Only positive antigen test results are verified by a RT-PCR. |
| Index test | Test review bias | The test of interest is called the index test, here being e.g. the antigen test. | The results of the index tests are interpreted with knowledge of the reference standard. | Antigen test results are interpreted with knowledge of the RT-PCR result. |
| | Threshold bias | | Diagnostic accuracy is evaluated for more than one threshold. | The threshold for the antigen test was not pre-specified. |
| Participant selection | Selection bias | Participant selection refers to the study population chosen for the respective study. | The study population is not representative of the population of the intended use. | Participants are selected based on results of RT-PCR. |
| | Spectrum bias | | Not the whole spectrum of the disease is included. | Cases are patients with severe symptoms, and controls are drawn from pre-pandemic samples. |

diagnostic tests and propose study designs to accelerate the evaluation of their diagnostic accuracy. Third, considering the challenges mentioned above, we propose a unified framework for rapid diagnostic test development and clinical evaluation. This highlights that multiple and perhaps different study designs will be necessary to build a convincing portfolio of evidence for various stakeholders during outbreaks of emerging infections.

## Diagnostic tests and testing strategies during the COVID-19 pandemic

For SARS-CoV-2, three types of tests can be distinguished according to their target: the polymerase chain reaction (PCR) test, the antigen test and the antibody test. The PCR test detects viral particles, the antigen test viral surface proteins and the antibody test SARS-CoV-2 specific antibodies. POC tests refer to those that can be evaluated directly on site. They are available for all three test types. While PCR and antibody tests are usually only performed by trained staff in hospitals and testing centers or similar, there are antigen tests for trained staff (rapid antigen test) but also as a home testing kit (antigen self-test, freely accessible)[7]. The cost of testing varied widely across phases of the pandemic, countries, type of test, and manufacturer. Rapid antigen tests now cost $1 in the United States of America (USA), PCR tests cost $5, and antibody tests cost $50 (test kit only with no personnel costs or similar; average approximate based on internet research and Du et al.[8]). PCR and antigen tests use nasal or throat swabs as specimen material, antibody tests use a blood sample. The PCR test takes up to 48 h to give results, whilst the antigen and the antibody test give results within 15 min. All tests are performed once, and a second test is often performed to confirm the test result (e.g., a PCR test to confirm a positive antigen test). A recently published network meta-analysis showed a mean sensitivity of 93% and specificity of 98% for PCR tests, 75% sensitivity and 99% specificity for antigen tests[9], and a Cochrane review reported a sensitivity and specificity of 94.3% and 99.8% for total antibody tests[10]. Throughout the pandemic, these tests were used in different combinations as part of various population-level testing strategies. Rapid antigen tests were used as part of screening and isolation programmes to detect asymptomatic infections in the community, and especially in key workers and workplaces. This was often combined with follow-up testing with PCR tests to minimise unnecessary isolation due to false positives.

Different testing strategies were used to fulfil various aims, e.g. full population screening programmes to break infection chains and studies such as the Real-time Assessment of Community Transmission (REACT) study for high-quality, real-time surveillance. Their strategies characteristics and costs differed accordingly. No public data on costs is available for most of these strategies, but modelling studies have been used to assess their cost-effectiveness under certain assumptions, taking the context in which the testing strategies were used into account and synthesizing all available evidence[8,11–13].

## Population-level information as a key input for public health decision-making

While diagnostic tests are usually developed for individual diagnosis and patient care, their results also play a crucial role in public health decision-making. Population-level case data, collected based on the number of positive diagnostic tests in surveillance systems worldwide, are a central input parameter for decision-making processes in public health policy. Cases might in this situation represent different outcomes of contact with an infectious agent (e.g., infections or deaths), and also different types of measures of this contact (e.g., incident or cumulative cases derived from seroprevalence studies).

Surveillance systems for infectious diseases provide reports on the number of cases associated with specific pathogens using standardized case definitions based on pre-defined rules (including diagnostic test results) and legal obligations. These surveillance systems run constantly for notifiable diseases associated with high public health risks[14]. Surveillance-related case data (based on diagnostic test results) are directly used for public health decision-making. They enable the development and parameterization of infectious disease models (e.g., for early warning and monitoring) and for

decision-analytic models (e.g., for assessing the benefit-harm, cost-effectiveness or other trade-offs when guiding public health interventions). This is especially true in epidemic or pandemic situations when reducing harm at a population level becomes a crucial aspect of the decision-making philosophy[15,16], and high consequence decisions must be made under uncertainty and time pressure. In such scenarios, two fundamental and extremely relevant quantities are some measure of the presence of the infection in the population (e.g., prevalence or incidence data) and a measure of existing immunity to the infection in the population, i.e. seroprevalence data.

An important decision supported by dynamic infectious disease modelling studies focusing on predicting infection dynamics is the timing of interventions. Interventions are most effective when deployed in time[17] and may cease to be effective if implemented too late[18]. Therefore, it is imperative that decisions about implementing interventions are made in a timely manner and sometimes with incomplete evidence, but with all relevant information being collected and reported appropriately. Monitoring population-level data from as soon as possible is essential since it can be used to set thresholds for starting interventions[19] and determine when intervention measures are no longer necessary and can be ended[20].

Due to reporting delays and the fact that the decision-making process is not instantaneous, decisions can come too late when relying solely on current population-level data. This is where infectious disease modelling comes in. Models help decision-makers obtain reasonable estimates of how the epidemic is likely to progress and what impact different interventions may have. This enables timely and informed decision-making[21–23]. Combined with benefit-harm and health economic models to account for unintended effects and costs of interventions, infectious disease models enable decision-makers to make optimal decisions given the available evidence and resources[24–26].

The points discussed above are exemplified by decision-making during the SARS-CoV-2 pandemic. Even during the early phases of the pandemic, decisions about interventions were made from population-level data. In the United Kingdom (UK), the timing of the first nationwide lockdown was determined based on the predicted number of people treated for SARS-CoV-2 in intensive care units (ICUs)[19]. In Australia, more targeted lockdowns were implemented based on regional prevalence data[27,28], and local lockdowns were also implemented in the UK during later phases of the pandemic[29]. Prevalence data became even more important when contact tracing and test-intervention strategies were implemented, because the predictive value of diagnostic tests depends on the infection prevalence. As vaccines became available, subpopulations most at risk of severe COVID-19 were prioritised and given the opportunity to be vaccinated first[30,31]. In Germany, vaccination and testing control rules for access to parts of public life varied from region to region. Again, the region-specific thresholds were based on the number of hospitalised patients testing positive for SARS-CoV-2 in the respective region[32].

Mathematical models were used throughout to support the decision-making process. The threshold for applying the first nationwide lockdown in the UK was set based on the number of people estimated to be potentially needing ICU treatment based on different modelling scenarios[19]. In Austria, the decision to prioritise vaccinating elderly and vulnerable groups was based on decision-analytic modelling aiming to minimise hospitalisations and deaths[33]. In general, infectious disease and decision-analytic models contributed substantially to the type and intensity of interventions implemented[34–36]. Once tests became widely available, they were also used to devise effective mass testing and isolating strategies[37,38].

The current pandemic has thus demonstrated the need for accurate and timely population-level case data and clinical case data (requiring different diagnostic tests and testing strategies), to allow public health policy decisions to be as well-informed as possible. Diagnostic tests, as the primary tool to obtain these population-level data, are therefore at the heart of all modelling efforts during an epidemic or pandemic, and early and precise knowledge about their accuracy is crucial for interpreting and further applying these case data.

## Challenges for diagnostic test evaluation in an epidemic setting
Diagnostic tests developed for emerging infections should serve various purposes, including individual clinical diagnosis, screening, and surveillance. These purposes demand distinct strategies and, in theory, require separate approval mechanisms[39]. However, test development, evaluation of technical validity, clinical validity and utility, as well as test validation currently do not account for the different uses in a generalized way. The challenges and potential solutions in this article and the framework proposed therein have been described with all these purposes in mind and are summarised in Box 1.

In the initial phase of an outbreak of an emerging infection, the main focus of diagnostic test development is providing a diagnostic test that can identify infected individuals with high sensitivity, so that they can be isolated and treated as soon as possible. This is especially important because the effectiveness of contact tracing depends directly on the quality and timeliness of case identification. Of course, a high specificity is also important, to prevent unnecessary isolation or treatment. This is usually achieved by direct detection of the pathogen, e.g., by molecular genetic tools such as PCR, microscopy, antigen tests or cultivation of the microorganisms involved. Later, a better understanding of the immune protection caused by contact with the agent is required, leading to the development of indirect pathogen detection tools such as antibody tests. Here, sensitivity and specificity are equally important to evaluate proxies of long-term immune protection and to detect past low severity infections which would have been missed otherwise. However, from the perspective of population-level modelling, some accuracy may be sacrificed if the true diagnostic accuracy of the test is known so that aggregate correction methods can be applied. Knowledge of the specificity of the direct detection tools developed earlier can also come into play in the case of reported reinfections, when it becomes important to understand whether these are true reinfections or due to false positives in a time of intensified testing. High specificity is also important once treatment options are available, but possibly come with relevant side effects, high costs or limited availability. Different population-level uses also require different diagnostic characteristics. Although PCR tests with a noticeable delay between testing and communication of results were used for population-level testing during the early phases of the COVID-19 pandemic, tests used for population screening generally need to be easily and quickly administered as POC tests, and lower accuracies, especially in specificity, are accepted as a trade-off for this. However, relatively high sensitivity is still important to make testing-and-isolation strategies effective. Deficiencies in specificity may be compensated for by confirmatory follow-up testing with highly specific tests to minimise unnecessary isolation. Furthermore, target populations, testing aims and prioritised estimators (e.g. sensitivity or specificity) can change rapidly, necessitating constant test evaluation and re-evaluation.

During an epidemic or pandemic, direct and indirect tests are thus used for different purposes and require different study designs, with different sample size calculations and study populations, to provide critical information with high precision and validity.

During epidemics with emerging infections, all new tests must, in general, quickly go through three steps: the test must be developed, its clinical performance assessed, and then information on its performance incorporated into infectious disease modelling to inform public health decision-making. Each step has potential sources of various biases that must be considered. Next we describe potential challenges during these steps and how these challenges might affect the submission process to regulatory agencies, also considering the perspective of test developers from industry.

## Diagnostic test development
Diagnostic tests for emerging infections typically are in the in vitro diagnostic (IVD) test category, as they examine human body specimens (e.g., nasopharyngeal swabs, nasal swabs, blood or saliva[39]). IVDs are generally considered medical devices[40]. Consequently, their development has to adhere to the rules of regulatory agencies and a pre-defined complex legal

## Box 1 | Summary of discussed challenges and proposed solutions

**There is a lack of samples from infected individuals for phase I test development studies during the early phases of an epidemic or pandemic.**
- Joint data collection and sharing infrastructure can be used to make infected samples available to test developers.

**Threshold selection for diagnostic tests must be done in advance, but the optimal threshold depends on ever-changing disease prevalences and consequences of misclassification.**
- A limited pool of promising thresholds can be evaluated simultaneously.
- Mixture modelling without defining a threshold can be used.
- Prevalence-specific thresholds can be developed and defined a-priori.

**Often-used two-gate designs for phase II studies likely lead to overestimation of diagnostic test accuracy.**
- Seamless enrichment designs, whereby proof-of-concept and confirmation are performed together as one study, can be used.

**Tests used as reference standards are themselves imperfect.**
- The use of follow-up data or composite reference standards that use all tests or clinical criteria available for diagnosis can alleviate this issue. However, one should be mindful of incorporation bias if the test under evaluation is part of the composite reference standard.

**The need for tests to be developed and evaluated rapidly to be used as part of public health interventions conflicts with the thorough study processes required to ensure transparency, reproducibility, and privacy.**
- Adaptive study designs can shorten time-to-market while maintaining the rigour of a high-quality study.

**Rapidly changing disease prevalences during recruitment may make a priori sample size calculations inappropriate.**
- Estimates of changing prevalence obtained via predictive modelling can be incorporated into the design of the study.

**Requirements such as sample size, properties of reference test, and inclusion criteria differ by country, increasing study complexity.**
- Careful upfront planning of multi-centre studies within a network such as the European Centre for Disease Control or a European Society of Clinical Microbiology and Infectious Diseases study group can keep complexity to a minimum.

**Diagnostic tests have different roles and target populations during different stages of an epidemic, requiring different performance characteristics and necessitating constant evaluation and re-evaluation.**
- Adaptive designs allow for sample size re-estimation and additional recruitment during the study to handle changing target populations.
- A longitudinal panel to be tested regularly using the test under evaluation can facilitate and expedite recruitment into diagnostic studies.
- A platform comparable to the REACT study or the ONS panel in the UK can be extended to make use of data from hospitals, health insurance companies, or public health agencies in diagnostic studies.
- Value-of-information analyses based on infectious disease modelling can help guide selection of optimal performance characteristics taking into account the purpose of the test being evaluated.

framework. Currently, the European Union (EU) IVD Regulation 2017/746 covers IVD medical devices, and focuses on a legislative process that prioritises individual safety, which means that different types of clinical data must be collected before submission. If a test is deemed capable of distinguishing infected individuals from non-infected ones, it has to be shown not to produce a one-off result[41].

There are several phase models for the development of diagnostic tests described in the literature. We discuss using the frequently used four-phase model[2,42,43]. The four phases for this are: I, evaluation of analytical performance; II, diagnostic accuracy estimation and determination of threshold; III, clinical performance estimation; and IV, evaluation together with diagnostic and/or therapeutic measures with regard to a patient-relevant endpoint.

Inter-rater agreement, analytical sensitivity (minimally detectable levels)[41] and cross-reactivity have to be investigated in the phase I studies to verify the technical validity, repeatability and reproducibility of laboratory tests (on a lot-to-lot, instrument group, and day-to-day basis). However, in the early phase of an epidemic or pandemic, there are often not enough samples from infected individuals. Sharing data and using a common infrastructure by, for instance, collecting samples at national reference centres, could solve this problem, if they are made accessible to IVD developers. A possible limitation of this approach is the risk of spectrum bias due to the particular mix of individuals, e.g. there may be more severe cases in the samples than in the population. Furthermore, regulatory agencies do not allow the use of (frozen) biobank samples for approval.

After having shown good technical performance, clinical performance in phase II and III studies must be demonstrated. An integral part of assessing the sensitivity and specificity of a continuous diagnostic test is the determination of the threshold at which it should be used[41]. This must be fixed before moving on to diagnostic test evaluation, to avoid bias caused by a data-driven threshold selection[44,45]. The optimal threshold for a diagnostic test depends on the prevalence and consequences associated with misclassification[38,46], which may both change over time. Thus a new study is needed every time the threshold changes, requiring extensive resources (particularly time and money).

Phase II studies are initial, so-called proof-of-concept studies covering clinical performance and are often carried out in a two-gate design[47], where sensitivity is estimated in diseased individuals and specificity in healthy samples from a different source. However, this design can lead to spectrum bias (Table 1). Sensitivity and specificity have been shown to be generally overestimated in such studies[47]. Likewise, a meta-analysis showed that a two-gate case-control design can lead to an overestimation of diagnostic accuracy[48]. In most situations outside an epidemic or pandemic, individuals tested are symptomatic and suspect they have the infection of interest, if the test is to be used to guide therapy or decide about isolation. However, during epidemics or pandemics, tested individuals can also be asymptomatic if the test is intended as a contact tracing or screening test[41]. In both cases, real-world samples may not be as perfect as in a laboratory situation[41] because testing can also be performed at POC, in the community, at the workplace, school, or home[39]. A test may require different performance characteristics if it is the first test in line, used to triage who will be tested further, compared to when the test is used to confirm infection. For instance, in a confirmation setting, most individuals who clearly do not have the infection of interest will be excluded[41].

### Diagnostic test evaluation

IVDs must be evaluated in phase III diagnostic accuracy studies that ideally start by including all individuals who will be tested in clinical practice to

avoid selection bias (all-comer studies). Individuals fulfilling the inclusion criteria should be enrolled consecutively, without judging how likely this person is to test positive or negative[41]. In such prospective diagnostic studies, to minimise variability and thus increase statistical power, all study participants ideally undergo all tests under investigation (index tests) as well as the reference standard to assign their final diagnosis.

The reference standard must be sufficiently reliable to differentiate between people with and without the target condition, but it is usually not perfect[41]. This imperfectness has to be taken into account when interpreting the results. Suppose a POC antigen test for SARS-CoV-2 is evaluated with a PCR test as reference standard resulting in a sensitivity of 90%. This does not mean that 90% of people with SARS-CoV-2 will be detected but that the POC test will be positive in 90% of cases with a positive PCR test. Solutions to this may include follow-up data or composite reference standards, which use all tests or clinical criteria available for a diagnosis. However, if the test under evaluation is part of this composite reference standard, this may lead to incorporation bias[49].

Depending on the phase of the epidemic or pandemic, recruitment speed can vary considerably due to changes in incidence. The guideline on clinical evaluation of diagnostic agents of the European Medicine Agency[2] demands sample size specification in a confirmatory diagnostic accuracy study in the study protocol. The required sample size is highly dependent on the prevalence of the target condition, which may change during the recruitment phase, making a priori sample size calculations inappropriate at the time of recruitment.

## Submission to regulatory agencies

Studies for industry face rigorous regulatory and ethics requirements as clinical trials follow strict processes and regulatory guidelines which are assessed in the regulatory submission process and are potentially controlled by audits. Clinical studies must be transparent, traceable and reproducible. Special attention must be paid to data quality and privacy. This leads to very detailed study preparation, documentation, quality control, and long and less flexible study processes.

When the SARS-CoV-2 pandemic began, the need for diagnostic tests grew with the rising number of cases. Regulatory bodies (like the U.S. Food & Drug Administration, FDA) established country-specific emergency use authorization guidelines[50,51] to make it easier and faster to bring a test for SARS-CoV-2 to the market and make it accessible during the pandemic. The WHO declared the end of the COVID-19 pandemic as a Public Health Emergency on 5 May 2023[52]. However, the FDA did not set a specific end date for the use of diagnostic tests authorized under Emergency Use Authorization (EUA), and they still remain valid under section 564 of the Federal Food, Drug, and Cosmetic Act, enabling uninterrupted use of EUA-authorized COVID-19 tests until further notice of regulatory transition requirements[53,54].

Submission process requirements such as sample size, inclusion criteria for subjects, and properties of the reference test differ between countries and may change during an epidemic or pandemic. Therefore, it is not always possible for a single study to be the basis for submissions to different countries or certificates, and several studies must be planned.

The different and changing requirements are not the only challenges submission teams face. The changing prevalence of infection makes adequate project management and timeline planning difficult. Recruitment of positive cases fulfilling the recruitment requirements can be slow which leads to a longer study duration and, subsequently, longer time to market. New mutations make re-evaluations of statistical properties necessary. Considering regulatory changes during pandemics and possible mutations, (pre)planning such a study is complicated and time-consuming.

## Potential solutions for the challenges presented

The challenges discussed previously are multidimensional but can be addressed by three countermeasures in several areas. First, test developers should use methodological approaches to address study designs and statistical analyses, increasing study efficiency and reducing the risk of bias. Second, strategic approaches and regulatory guidance for the industry should be deployed to clearly define opportunities but also limitations in the development and approval process. Third, results and feedback from population-level mathematical modelling should inform test development and validation for deriving optimal study designs based on formal value-of-information analyses.

**Methodological solutions.** Methodological solutions fall into two categories; statistical methods to control bias, and those to increase speed and efficiency.

The different biases in diagnostic studies have been described extensively, both in general[55–57] and also specifically in the context of the SARS-CoV-2 pandemic[58] and POC tests for respiratory pathogens[59]. From a methodological standpoint, the problem of bias can be addressed in two ways: either by choosing a study design in the planning stage that minimises the risk of bias, or by using analytical methods that correct for potential bias.

An excellent overview of how to avoid bias through an appropriate design can be found in Pavlou et al.[60]. Important for the planning phase is the work of Shan et al.[61], who present an approach to calculate the sample size in the presence of verification bias (i.e., partial or differential verification bias).

In terms of bias reduction methods during the analysis phase, most studies focus on the correction of verification bias. Bayesian approaches are mainly proposed for differential verification bias[62,63], while there are a variety of methods for partial verification bias[64].

Time to market has to be reduced significantly in pandemics to find an optimal trade-off between misclassification and missed opportunities for action. From a statistical point of view, the methods and processes must be reconsidered. One possibility to improve study designs and statistical analysis is adaptive designs, that can increase efficiency. These approaches have been long established in therapeutic studies and are also anchored in guidelines[3,65]. With adaptive designs, it is possible to make pre-specified modifications during the study. For example, inclusion and exclusion criteria can be changed, the trial can be terminated early due to futility or efficacy, or the sample size can be recalculated. The characteristics and typical adaptive designs have been very clearly summarised[66]. A review of published studies with adaptive designs showed that the pharmaceutical industry in particular increasingly uses simple adaptive designs, with more complex adaptive designs still being rare[67].

In diagnostic studies, however, experience in using adaptive designs in diagnostic clinical trials for submissions is limited. A summary of the current state of research is available for diagnostic accuracy studies[68], for randomised test-treatment studies[69] and for adaptive seamless designs[70]. Methods for blinded and unblinded sample size re-calculations for diagnostic accuracy studies have been published recently[71–73], as well as adaptive designs for test-treatment studies[74] and adaptive seamless designs. The diagnostic industry heavily depends on regulatory guidelines worldwide. If regulatory bodies emphasise more efficient diagnostic trials that include, e.g., adaptive designs, the implementation of modern study designs will be incentivised.

In the following, concrete possible solutions to the above-mentioned challenges are explained as examples. For details, please refer to the corresponding articles.

Firstly, the problem of setting a threshold in an early study that may later turn out not to be optimal can be addressed by selecting a limited pool of promising thresholds[75]. These are then evaluated simultaneously in the validation study, with the type I error adjusted accordingly. Another approach is to use mixture modelling without defining a threshold[76]. Prevalence-specific cut-offs might be developed and defined a priori.

Secondly, if the testing strategy and thus the target population change during the study, adaptive designs offer the possibility to re-estimate the sample size in a blinded manner based on the prevalence estimated in the interim analysis[71].

Thirdly, a seamless enrichment design can be chosen to address the problem of biased diagnostic accuracy in two-gate designs, in which proof-

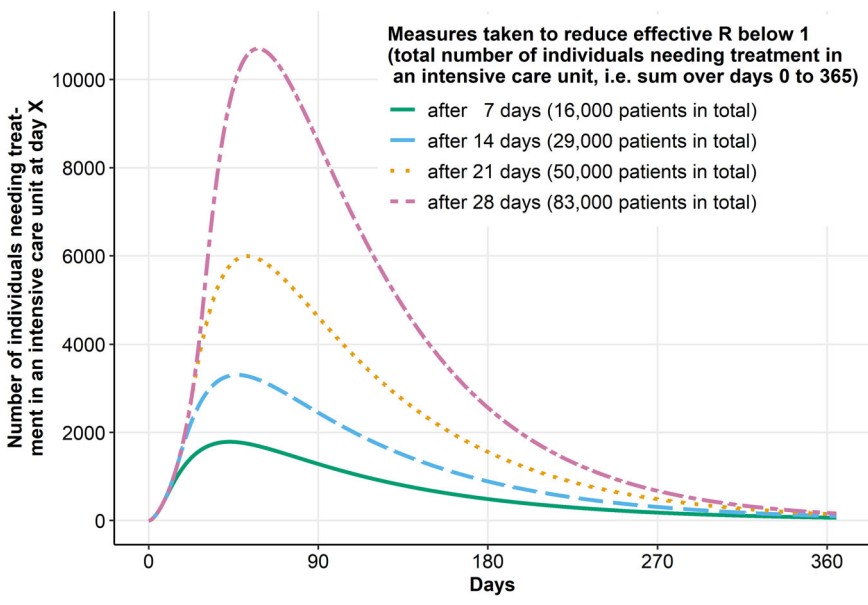

**Fig. 1 | Model-based demonstration of effect of delays in implementing public health interventions on number of individuals needing treatment during an epidemic.** Time course of the number of individuals requiring intensive care therapy if measures are taken at different time points to reduce the effective reproduction number below 1 (reproduction number (R) at the start set to R = 2, reduced to R = 0.9 after 7, 14, 21, 28 days). Assumptions of the Susceptible-Exposed-Infectious-Recovered (SEIR) model: pre-infection time: 3 days, infectious time: 7 days, proportion of individuals requiring intensive care unit out of all infectious patients: 2%, length of stay in intensive care unit: 15 days, population size: 80.000.000, number of susceptible individuals at the start: 79.950.000, number of exposed individuals at start: 40.000, number of infectious individuals at start: 10.000, number of immune individuals at the start: 0, number of individuals requiring intensive care at the start: 0.

of-concept and confirmation are performed together in one study. However, it is apparent that regulatory authorities are cautious of the possible short-comings of these innovative designs, and a lot of work needs to be done to get them approved[77]. This, in turn, results in the manufacturers of diagnostic tests being conservative in their study designs.

**Solutions for political decision-making based on mathematical modelling.** The accuracy, accessibility, and costs of diagnostic tests all play a role in decisions about testing programmes, and the decision on whether, for instance, a more sensitive test with lower accessibility and higher costs (in the context of SARS-CoV-2, e.g. a real-time PCR test) should be administered with low frequency or a less sensitive test with better accessibility and lower costs (in the context of SARS-CoV-2, e.g. a rapid antigen test) should be administered with higher frequency is context-specific. Mathematical models can and should support these decisions in real-time, as was the case during the COVID-19 pandemic[78].

When considering model input data, one key aspect that has to be taken into account by modelling studies is the deliberate parameterization of accuracy for case numbers based directly or indirectly on the results of a diagnostic test[79]. This typically includes incidence rates as well as seroprevalence estimates. Knowledge about the diagnostic accuracy of the tests is critical as biased estimates of sensitivity and specificity can lead to biased estimates in modelling results used for health decision making. The textbook example for this is an overestimation of the specificity of an antibody test used for seroprevalence studies in low-prevalence settings which leads to an overestimation of the proportion of the population which has had already contact with the emerging infectious agent. As a consequence, population immunity would be overestimated, underestimating the risk associated with an uncontrolled spread in the population. If true diagnostic accuracy is known, population-level estimates on e.g. seroprevalence can be corrected, either before the modelling study or within the modelling framework[80]. If it is not known that the proposed diagnostic accuracy is biased, modelling can help in detecting implausibilities, especially if parameter fitting is carried out regularly. Here, modelling can inform diagnostic test evaluation with respect to potential biases, but also in the context of value of information analyses. During the first three months of the SARS-CoV-2 pandemic, only a minority of modelling studies in the field accounted for test accuracy estimates; the remaining used incidence and later seroprevalence data as if they represented the ground truth. This approach would be appropriate if incidence or seroprevalence data were already corrected for imperfect test accuracy estimates. However, in this case, the correction procedure should

still be reported in the modelling study to enable a transparent evaluation of model parameterization, and the model(s) should be reparametrized once updated information on diagnostic test accuracy is available. Earlier decision making based on updated information increases the impact of these decisions on population health (Fig. 1). Decisions just a few weeks or even a couple of days earlier can make a huge difference, offering a critical time window for accelerated diagnostic studies. Fig. 2 shows the sensitivity of model-based assessments of interventions to diagnostic test accuracy parameters. The results show that even relatively small biases in the estimation of test accuracy (much smaller than those found in the Cochrane reviews) for an antibody test used to derive the proportion of undetected cases in a population have an enormous effect on the predicted further course of the epidemic (the mechanism for this impact is that the proportion of undetected cases is used to correct reported case numbers before they are used to calibrate transmissibility estimates and other parameters). The results are enough to change public health decision-making from, for example, not implementing population-level contact reduction measures to introducing a hard lockdown if the defined outcome of interest crosses a set decision-analytic threshold.

**Longitudinal panels as a platform for diagnostic accuracy studies.** Given the rapidly changing research questions during an epidemic or pandemic, there is a huge practical challenge in setting up diagnostic studies even with the modern study designs described above, because the acceptable time spans for recruiting study participants and for conducting the actual studies are very short. The availability of a study platform that allows immediate initiation of diagnostic studies reflecting the current research question and infection dynamics is indispensable for timely studies in the field. One way to ensure this is the sustainable implementation of a longitudinal panel within existing cohorts (e.g., as the NAKO Health Study[81]) that is tested regularly for the presence or absence of the pathogen by a defined test (or several) under evaluation. Another approach is to use data from hospitals, health insurance or public health agencies. For example, a platform comparable to the UK Office for National Statistics (ONS) panel[82] or the REACT study[83] can be built and used to evaluate the tests or testing strategies under study, and for real-time communication of the results of the respective tests representing current or past infection dynamics. In this setting, flexible and fast study designs can fulfil both, equally important, purposes at the same time.

**Feedback triangle at the centre of a unified framework.** As discussed above, the development and evaluation of diagnostic tests in an epidemic

**Fig. 2 | Example model forecasts of number of individuals needing treatment during an epidemic, under varying assumptions about antibody test specificity during parameterisation.** Forecast for number of individuals requiring intensive care therapy if measures as described for Fig. 1 are taken after 28 days, with the model parametrised using different assumptions for antibody test specificity. SEIR model with Infectious compartment split into 'Detected' and 'Undetected' compartments. Parametrisation as described in Fig. 1, but with 2% of *detected* infectious cases requiring intensive care. Proportion of detection obtained from data from the 'Heinsberg' seroprevalence study[86], corrected for antibody test sensitivity of 0.9, and specificity of 0.9, 0.93, 0.96, 0.99, 1.

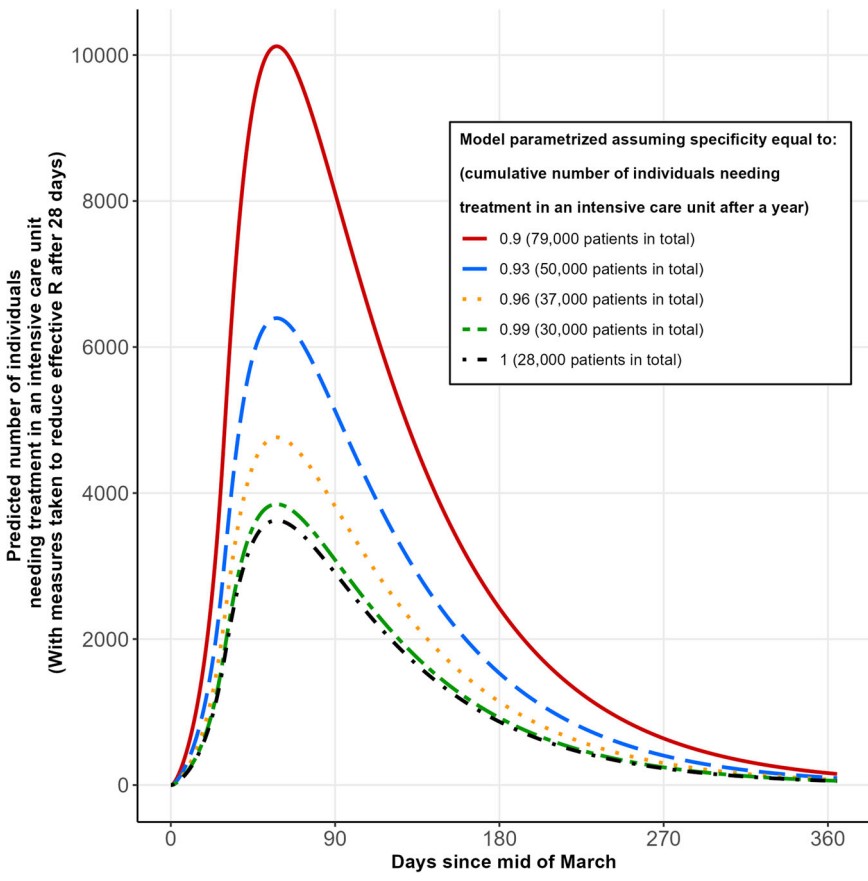

**Fig. 3 | Feedback loop at core of proposed framework.** Schematic representation of the feedback triangle (in green) between diagnostic test accuracy results, the parametrization of modelling studies and their results, the consequences for decision analysis, and the test strategy chosen based on these decisions.

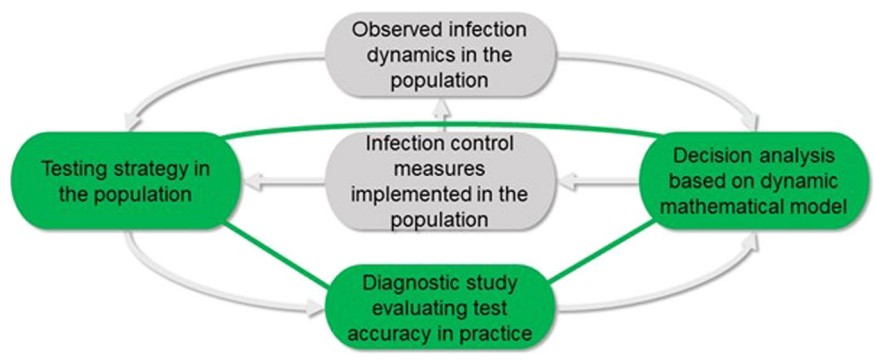

or pandemic setting is closely linked to modelling studies used to inform political and public health decision-making. This link is at the centre of the unified framework we propose based on experiences during the SARS-CoV-2 pandemic (Fig. 3). The execution of diagnostic studies for new tests or new application areas of existing tests depends heavily on current test strategies and those potentially applied in the future. Results from diagnostic studies are a direct input in mathematical modelling studies, and in turn results from these models are used for decision-making based on a defined decision-making framework. However, modelling studies can also give crucial feedback to those responsible for planning and analysing diagnostic accuracy studies. Here, so-called value-of-information analyses can help identify those gaps in knowledge regarding diagnostic test accuracy that need to be tackled first or require the greatest attention[84]. This can directly affect sample size estimations, for instance if more precision is needed to estimate the test's specificity (as

is often the case with antibody tests). Therefore, the optimal strategy to deal with these constant feedback loops is to establish continuous collaboration between the disciplines representing the three parts of this loop (in green in Fig. 3). This collaboration platform can use the longitudinal panel with complementary perspectives described above to create a unified diagnostic test development and evaluation framework during an epidemic or pandemic. The modern study designs and bias reduction methods described above can be applied to obtain the best potentially available evidence about diagnostic test accuracy in different settings. When creating such a framework for developing and evaluating diagnostic tests and considering the corresponding results in modelling studies, both infection-specific (e.g. transmission rates, case fatality ratios) and test-specific characteristics (e.g. test type, costs, availability) must be considered, especially for the collection of population-level data from testing programmes.

**Diagnostic test-intervention studies using a cluster-randomised approach.** In many situations, diagnostic test accuracy estimates should only be seen as surrogate information since the actual outcome of interest during an ever-changing pandemic, especially in the later phases, is the effect of an application of this test on clinical or population-level outcomes. Here, it is possible, as has been discussed during the SARS-CoV-2 pandemic, to take a step further and move test evaluation to phase IV or diagnostic test-intervention studies. In this phase, individuals or clusters of individuals are randomised to a diagnostic strategy (e.g., regular testing of the entire population or testing only in case of symptoms). The relevant clinical endpoint is then compared between randomised groups[42]. Thus, the test strategy is treated as an intervention evaluated for its effectiveness and safety. Diagnostic test accuracy helps to reach this endpoint but is not the only factor under evaluation. The practicability of the strategy, as well as real-world effectiveness and interaction with other interventions (e.g., the case isolation and quarantine of close contacts), are also assessed indirectly in this approach. In a dynamic infectious disease setting where an intervention can have indirect effects on people other than the target population, only cluster-randomised approaches allow for a reasonable estimation of population-level effects of the intervention under study. In infectious disease epidemiology, similar designs are applied when assessing the effectiveness of vaccination programs on a population level, often combined with a staggered entry approach to allow all clusters to benefit from the intervention over time (so-called stepped-wedge design). During the pandemic, small-scale pilot studies were discussed, trying to mirror such an approach in a non-randomised way, often claiming to be a natural experiment. However, most of them did not follow guidelines and recommendations available for diagnostic test-intervention studies that would have improved the quality of the results and their usefulness for evidence-based public health. Rigorous application of cluster-randomised diagnostic test-intervention studies to implement testing strategies can support decision-making processes in the later stages of an epidemic or pandemic.

## Conclusion

The development and evaluation of diagnostic tests for emerging infectious agents during an epidemic or pandemic come with many serious challenges. We propose integrating diagnostic studies in a unified framework representing the triangle of diagnostic test evaluation, predictive or decision-analytic public health modelling and the testing strategy applied in this population. This framework can use modern, flexible and fast study designs and should incorporate a longitudinal panel as a continuous study platform. Diagnostic test-intervention studies need to be planned early and should be used for evidence-based public health during later phases of an epidemic or pandemic, when research questions become more complicated and testing strategies serve as interventions to counteract infectious disease dynamics.

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

## Acknowledgements
This work was supported by unrestricted grants from the Deutsche Forschungsgemeinschaft (German Research Council, grant numbers KA 5361/1-1 and ZA 687/3-1, project number 458526380).

## Author contributions
M.C. contributed to writing the first draft, attended the workshop and participated in the discussions, and contributed to and approved the final version of the manuscript. D.K. contributed to writing the first draft, attended the workshop and participated in the discussions, and contributed to and approved the final version of the manuscript. P.B. attended the workshop, participated in the discussions, and contributed to and approved the final version of the manuscript. O.G. attended the workshop, participated in the discussions, and contributed to and approved the final version of the manuscript. A.J. attended the workshop, participated in the discussions, and contributed to and approved the final version of the manuscript. M.K. attended the workshop, participated in the discussions, and contributed to and approved the final version of the manuscript. M.L. participated in the discussions and contributed to and approved the final version of the manuscript. R.M. attended the workshop, participated in the discussions, and contributed to and approved the final version of the manuscript. J.R. attended the workshop, participated in the discussions, and contributed to and approved the final version of the manuscript. N.S.-M. attended the workshop, participated in the discussions, and contributed to and approved the final version of the manuscript. U.S. participated in the discussions and contributed to and approved the final version of the manuscript. C.S. participated in the discussions and contributed to and approved the final version of the manuscript. C.E. participated in the discussions and contributed to and approved the final version of the manuscript. N.R. contributed to writing the first draft, attended the workshop and participated in the discussions, and contributed to and approved the final version of the manuscript. A.K. contributed to writing the first draft, attended the workshop and participated in the discussions, and contributed to and approved the final version of the manuscript. A.Z. contributed to writing the first draft, attended the workshop and participated in the discussions, and contributed to and approved the final version of the manuscript.

## Funding

## Competing interests
The authors declare no competing interests.

## Additional information

[1]Institute of Epidemiology and Social Medicine, University of Münster, Münster, Germany. [2]Institute of Medical Biometry and Epidemiology, University Medical Center Hamburg-Eppendorf, Hamburg, Germany. [3]Amsterdam University Medical Centers, University of Amsterdam, Epidemiology and Data Science, Amsterdam, The Netherlands. [4]Department of Clinical Research, University of Southern Denmark, Odense, Denmark. [5]Department of Infectious Disease Epidemiology, NRW Centre for Health, Bochum, Germany. [6]Julius Center for Health Sciences and Primary Care, University Medical Center Utrecht, Utrecht University, Utrecht, The Netherlands. [7]Institute of Medical Microbiology, Virology and Hygiene, University Medical Center Hamburg-Eppendorf, Hamburg, Germany. [8]Institute for Medical Epidemiology, Biometrics and Informatics, Interdisciplinary Center for Health Sciences, Medical Faculty of the Martin Luther University Halle-Wittenberg, Halle, Germany. [9]NMI Natural and Medical Sciences Institute at the University of Tübingen, Reutlingen, Germany. [10]Department of Public Health, Health Services Research and Health Technology Assessment, Institute of Public Health, Medical Decision Making and Health Technology Assessment, UMIT- University for Health Sciences, Medical Informatics and Technology, Hall in Tirol, Austria. [11]Division of Health Technology Assessment and Bioinformatics, ONCOTYROL - Center for Personalized Cancer Medicine, Innsbruck, Austria. [12]Center for Health Decision Science, Departments of Epidemiology and Health Policy & Management, Harvard T.H. Chan School of Public Health, Boston, MA, USA. [13]Program on Cardiovascular Research, Institute for Technology Assessment and Department of Radiology, Massachusetts General Hospital, Harvard Medical School, Boston, MA, USA. [14]Roche Diagnostics GmbH, Penzberg, Germany. [15]These authors contributed equally: Madhav Chaturvedi, Denise Köster. [16]These authors jointly supervised this work: Nicole Rübsamen, André Karch, Antonia Zapf. ✉e-mail: andre.karch@ukmuenster.de

