## [Transparent Peer Review file · Communications Medicine]

A unified framework for diagnostic test development and evaluation during outbreaks of emerging infections

Corresponding Author: Dr André Karch

Version 0:

Reviewer comments:

Reviewer #1

(Remarks to the Author)

This paper summarized the discussions on diagnostic test development and evaluation in a workshop in the context of COVID-19 pandemic. The manuscript is clear and well-written, though I found I might not have the expertise to comment on all the aspects discussed regarding test development and evaluation. Please find below my suggestions or comments for consideration.

1. I agree with the authors that population-level information from testing programmes is important for health decision making. However, I have reservation to include that as one of the evaluation criteria for diagnostic tests. The primary goal for diagnostic tests is to detect infection accurately. However, to obtain accurate information of infection incidence or prevalence for mathematical modelling, it also requires that the tests are easily accessible (e.g., PCR tests vs. rapid antigen tests) and that the testing programmes are well designed (e.g., symptom-based testing vs. population-based sampling in surveillance programmes such as the REACT study in the UK). I suggest the authors clearly differentiate the goals and aims of the evaluation of diagnostic tests serving different purposes.

2. The authors did mention that the phrase or stage of the pandemic matters and it is one of the challenges of the diagnostic test evaluation. I suggest expanding the discussion with more context and how it is related to the sensitivity and specificity of the tests. For example, in the early stage of the pandemic while contact tracing and case finding was important, both sensitivity and specificity matter a lot; while in the later stage of the pandemic, testing programmes are mainly for the purpose of surveillance which means that tests with compromised sensitivity (such as rapid antigen tests) might be useful as long as the sensitivity and specificity of the tests are well characterized.

3. It would be great to add some discussion on how bias might be generated from results of tests with compromised sensitivity or specificity and how these bias might be adjusted in mathematical models to inform health decision making.

4. Finally, it would be great to add discussion about access and frequency of testing. For example, to serve purposes of different testing programmes, which combinations would be preferred:

- 1) More sensitive test + Less frequent testing + Not easily accessible (e.g., RT-PCR)
- 2) Less sensitive test + More frequent testing + Easily accessible (e.g., rapid antigen test)

5. Costs and cost-effectiveness were briefly mentioned. It would be great to compare the costs and cost-effectiveness of the testing programmes mentioned in the manuscript, e.g. the symptom-based testing programmes used in many countries and the REACT study from the UK.

Reviewer #2

(Remarks to the Author)

The paper describes challenges and solutions related to diagnostic tests during epidemics. Can you include a table that includes major challenges and solutions so that we can easily understand the ideas?

Reviewer #3

(Remarks to the Author)

The present paper describes the output from a multidisciplinary workshop on a framework for diagnostic test development and evaluation in an epidemic or pandemic situation. This framework considers a feedback loop between diagnostic accuracy studies and modelling studies for public health decision making and public health interventions.

In my opinion, the paper succeeded in suggesting a unified framework with a feedback triangle between diagnostic test accuracy results, parameterization for predictive and decision-analysis modelling studies, and population test strategy selected based on this analysis.

The authors suggest concrete solutions for the challenges of diagnostic accuracy studies in a pandemic situation, such as the changing thresholds for a diagnostic test, changing testing strategies and target populations, and the presence of biased designs. These are the use of adaptive study designs, which address bias, in addition to the use of analytical methods. In conclusion, I agree with the authors that it is very important to use a longitudinal panel as a continual study platform.

I strongly support the publication of the paper because of my experience as a medical doctor trained in clinical microbiology and infectious disease epidemiology.

Minor comments: changing table; see uploaded table.

Version 1:

Reviewer comments:

Reviewer #1

(Remarks to the Author)

The authors have addressed all my comments and I don't have further comments on the manuscript.

Reviewer #2

(Remarks to the Author)

The authors have addressed the comments, and it can be published as the current form.

Dear reviewers,

We thank you for the time and effort spent evaluating our manuscript “*A unified framework towards diagnostic test development and evaluation during outbreaks of emerging infections*” and for your constructive comments and suggestions for improvement. We have revised our manuscript accordingly, and hope that we have adequately addressed your comments. Below are details about how we have addressed each reviewer comment.

Reviewer #1

1. I agree with the authors that population-level information from testing programmes is important for health decision making. However, I have reservation to include that as one of the evaluation criteria for diagnostic tests. The primary goal for diagnostic tests is to detect infection accurately. However, to obtain accurate information of infection incidence or prevalence for mathematical modelling, it also requires that the tests are easily accessible (e.g., PCR tests vs. rapid antigen tests) and that the testing programmes are well designed (e.g., symptom-based testing vs. population-based sampling in surveillance programmes such as the REACT study in the UK). I suggest the authors clearly differentiate the goals and aims of the evaluation of diagnostic tests serving different purposes.

Thank you for the comment. We agree that the entire topic is complex, and that arguing that population-level consequences need to be taken into account already at the stage of test development or early evaluation is not necessarily helpful. We now make clear that we don't want to push into this direction, and comment on additional dimensions (like availability and price) to be considered when having population-level effects in mind. We have also added a section with an overview of the tests used in the context of COVID-19, and how they relate to these points (lines 110-127):

“For SARS-CoV-2, three types of tests can be distinguished according to their target structure: the PCR test, the antigen test and the antibody test. The PCR test detects viral particles, the antigen test viral surface proteins and the antibody test SARS-CoV-2 specific antibodies. Point of care (PoC) tests refer to those that can be evaluated directly on site, and are available for all three test types. While PCR and antibody tests are usually only performed by trained staff in hospitals and testing centers or similar, there are antigen tests for trained staff (rapid antigen test) but also as a home testing kit (antigen self-test, freely accessible)⁸. The cost of testing varies widely between phases of the pandemic, countries, type of test, and manufacturer. Rapid antigen tests now cost \$1 in the USA, PCR tests cost \$5, and antibody tests cost \$50 (test kit only with no personnel costs or similar; average approximate based on internet research and Du et al⁹). PCR and antigen tests use nasal or throat swabs as specimen material, antibody tests use a blood sample. The PCR test leads within up to 48 hours to results, the antigen and

the antibody test within up to 15 minutes. All tests are performed once, and a second test is often performed to confirm the test result (e.g., a PCR test to confirm a positive antigen test). A recently published network meta-analysis showed a mean sensitivity of 93% and specificity of 98% for PCR tests, 75% sensitivity and 99% specificity for antigen tests¹⁰, and a Cochrane review reported a sensitivity and specificity of 94.3% and 99.8% for total antibodies¹¹.

AND (lines 515-520)

“When creating such a framework for developing and evaluating diagnostic tests and considering the corresponding results in modelling studies, both infection-specific (e.g. transmission rates, case fatality ratios) and test-specific characteristics (e.g. test type, costs, availability) must be taken into account, especially for the collection of population-level data from testing programmes”

2. The authors did mention that the phrase or stage of the pandemic matters and it is one of the challenges of the diagnostic test evaluation. I suggest expanding the discussion with more context and how it is related to the sensitivity and specificity of the tests. For example, in the early stage of the pandemic while contact tracing and case finding was important, both sensitivity and specificity matter a lot; while in the later stage of the pandemic, testing programmes are mainly for the purpose of surveillance which means that tests with compromised sensitivity (such as rapid antigen tests) might be useful as long as the sensitivity and specificity of the tests are well characterized.

The discussion on this has now been elaborated on, and reads as follows (lines 212-240):

“In the initial phase of an outbreak of an emerging infection, the main focus of diagnostic test development is providing a diagnostic test that can identify infected individuals with high sensitivity, so that they can be isolated and treated as soon as possible; this is especially important because the effectiveness of contact tracing depends directly on the quality and timeliness of case ascertainment. Of course, a high specificity is also important, to prevent unnecessary isolation or treatment. This is usually achieved by direct detection of the pathogen, e.g., by molecular genetic tools like polymerase chain reaction (PCR), microscopy, antigen tests or cultivation of the microorganisms involved. Later, a better understanding of the immune protection caused by contact with the agent is required, leading to the development of indirect pathogen detection tools antibody tests. Here, sensitivity and specificity are equally important to evaluate proxies of long-term immune protection and to detect past low severity infections which would have been missed otherwise; however, from the perspective of population-level modelling, some accuracy may be sacrificed as long as the true diagnostic accuracy of the test is known so that aggregate correction methods can be applied. Knowledge of the specificity of the direct detection tools developed earlier can also come into play in the case of reported reinfections,

whereby it becomes important to understand whether these are true reinfections or are due to false positives in a time of intensified testing. High specificity is also important once treatment options are available, but possibly come with relevant side effects, high costs or limited availability. Different population-level uses also require different diagnostic characteristics. Although PCR tests with a noticeable delay between testing and communication of results were used for population-level testing during the early phases of the COVID-19 pandemic, tests used for population screening generally need to be easily and quickly administered as point-of-care (POC) tests, and lower accuracies, especially in specificity, are accepted as a trade-off for this; relatively high sensitivity is still important here to make testing-and-isolation strategies effective. Deficiencies in specificity may be made up by confirmatory follow-up testing with highly specific tests to minimise unnecessary isolation. Furthermore, target populations, testing aims and prioritised estimators (e.g. sensitivity or specificity) can change rapidly, necessitating constant test evaluation and re-evaluation.”

Furthermore, we have addressed this issue in the section *Feedback triangle at the centre of a unified framework* as follows (lines 515-520, also referred to for the point above):

“When creating such a framework for developing and evaluating diagnostic tests and considering the corresponding results in modelling studies, infection-specific and test-specific characteristics must be taken into account. Infection-specific characteristics are e.g. transmission rate or case fatality ratio, test-specific characteristics are e.g. test type (PCR, antigen, or antibody) or costs and availability of the test, especially for the collection of population-level data from testing programmes.”

3. It would be great to add some discussion on how bias might be generated from results of tests with compromised sensitivity or specificity and how these bias might be adjusted in mathematical models to inform health decision making.

Thank you for this suggestion. We added a paragraph on this (lines 425-437):

“Knowledge about the diagnostic accuracy of the respective tests is critical as biased estimates of sensitivity and specificity can lead to biased estimates in modelling results used for health decision making. The textbook example for this is an overestimation of the specificity of an antibody test used for seroprevalence studies in low-prevalence settings which leads to an overestimation of the proportion of the population which has had already contact with the emerging infectious agent. As a consequence, population immunity would be overestimated, underestimating the risk associated with an uncontrolled spread in the population. If true diagnostic accuracy is known, population-level estimates on e.g. seroprevalence can be easily corrected, either before the modelling study or within the modelling framework. If it is not known that the proposed diagnostic accuracy is biased, modelling

can help in detecting implausibilities—especially if continuous parameter fitting takes place. Here modelling can inform diagnostic test evaluation with respect to potential biases, but also in the context of value of information analyses.”

4. Finally, it would be great to add discussion about access and frequency of testing. For example, to serve purposes of different testing programmes, which combinations would be preferred:

1) More sensitive test + Less frequent testing + Not easily accessible (e.g., RT-PCR)

2) Less sensitive test + More frequent testing + Easily accessible (e.g., rapid antigen test)

Thank you for bringing up this important point. Since decisions like this are context-specific and depend of course very much on the multidimensional aspects discussed above and in the manuscript, we decided to discuss this issue on an abstract level, using your example as an illustration (lines 414-421):

“The accuracy, accessibility, and costs of diagnostic tests all play a role in decisions about testing programmes, and the decision on whether, for instance, a more sensitive test with lower accessibility and higher costs (in the context of SARS-CoV-2, e.g. a real-time PCR test) should be administered with low frequency or a less sensitive test with better accessibility and lower costs (in the context of SARS-CoV-2, e.g. a rapid antigen test) should be administered with higher frequency is context-specific. Mathematical models can and should support these decisions in real-time, as was the case during the COVID-19 pandemic⁷⁸.”

5. Costs and cost-effectiveness were briefly mentioned. It would be great to compare the costs and cost-effectiveness of the testing programmes mentioned in the manuscript, e.g. the symptom-based testing programmes used in many countries and the REACT study from the UK.

This is of course a very critical point, which is in its entirety unfortunately beyond the scope of our manuscript as the respective costs e.g. of the REACT study are not publically available and a cost-effectiveness analysis as such not possible. There are, however, modelling studies estimating the cost-effectiveness of different strategies dependent on various factors, as e.g. prevalence and disease burden. We added a paragraph on this (lines 132-137):

“Different testing strategies were used to fulfil various aims, e.g. full population screening programmes to break infection chains and studies like the REACT study for high-quality, real-time surveillance, and their characteristics and costs differed accordingly. No public data on costs is available for most of these strategies, but modelling studies have been used to assess their cost-effectiveness under certain assumptions, taking the context in which the testing strategies were used into account and synthesizing all available evidence^{9,12–14}.”

Reviewer #2

The paper describes challenges and solutions related to diagnostic tests during epidemics. Can you include a table that includes major challenges and solutions so that we can easily understand the ideas?

We have now included the table below as suggested (Table 2 in the manuscript)

Challenge	Proposed Solution
Lack of samples from infected individuals for phase I test development studies during the early phases of an epidemic or pandemic.	• Joint data collection and sharing infrastructure e.g. sample collection at national reference centres can be used to make infected samples available to test developers.
Threshold selection for diagnostic tests must be done in advance during development, but optimal threshold depends on disease prevalence and consequences of misclassification; these change over time, requiring new studies every time the threshold changes.	• A (limited) pool of promising thresholds can be selected, to be evaluated simultaneously.• Mixture modelling without defining a threshold can be used.• Prevalence-specific cut-offs can be developed and defined a priori.
Diagnostic tests have different roles, requiring different performance characteristics, and different target populations during different stages of an epidemic; this necessitates constant test evaluation and re-evaluation, using various different study designs. Mutations in the disease-causing pathogen also necessitate re-evaluation.	• Adaptive designs, long in use in therapeutic studies but still a developing field in diagnostic test studies, allow for sample size re-estimation and additional recruitment during the study and can thus handle changing target populations.• Implementing a longitudinal panel within existing cohorts to be tested regularly for the presence of the relevant pathogen using the test under evaluation can facilitate and expedite recruitment in diagnostic studies and make conducting multiple studies easier and faster.• A platform comparable to the REACT study or the ONS panel in the UK can be extended to make use of data from hospitals, health insurance companies, or public health agencies in diagnostic studies.• Value-of-information analyses based on infectious disease modelling can help guide selection of optimal performance characteristics that take into account the purpose of the test being evaluated.
Often-used two-gate designs for phase II studies are likely to lead to overestimation of diagnostic test accuracy.	• Seamless enrichment designs, whereby proof-of-concept and confirmation are performed together as one study, can be used.

Tests used as reference standards during diagnostic test evaluation are themselves imperfect.	 • The use of follow-up data, or composite reference standards that use all tests or clinical criteria available for a diagnosis, can alleviate this issue, although one should be mindful of incorporation bias if the test under evaluation is part of the composite reference standard.
In an epidemic or pandemic, disease prevalences may change rapidly during the recruitment phase of diagnostic test studies, making a priori sample size calculations inappropriate.	 • Estimates about the changing prevalence obtained from infectious disease modelling can be incorporated into the design of the diagnostic test study.
Requirements like sample size, properties of reference test, and inclusion criteria differ by country, meaning several different studies must be planned.	 • Careful, upfront planning of studies, e.g. by using different sample sizes for different countries, including different patients for different target countries, and including more than one reference test at the labs, can keep complexity to a minimum. Planning multi-centre studies within a professional network like the eCDC or an ECCMID study group can facilitate this.
The need for tests to be developed and evaluated rapidly to be used as part of public health interventions during an epi-/pandemic conflicts with the detailed and long study processes required to ensure transparency, reproducibility, and privacy.	 • Adaptive designs can be used to shorten time-to-market while maintaining the rigour required of a high-quality study.

Reviewer #3

Minor comments: changing table; see uploaded table.

Thank you for this hint. We have restructured the table as suggested.

Domain	Definition of domain	Type of bias	Description	Example based on the SARS-CoV-2 antigen test
Reference standard	“This is the test used to define the target condition, and the underlying assumption is that it reflects the truth. By design, the reference standard is assumed to be flawless. The reference standard sets the reference, and sensitivity and specificity are expressed as the proportion of reference standard positives with a positive index test result, and the proportion of reference standard negatives with a negative index test result, respectively.”⁷ For the evaluation of the SARS-CoV-2 antigen test, the detection of viral RNA via reverse transcription–polymerase chain reaction (RT-PCR) is in general the reference standard. This might be problematic if the target condition is e.g. being infectious.	Imperfect gold/reference standard bias	The reference standard does not correctly classify the target condition.	A single RT-PCR is used as a reference standard.
		Incorporation bias	The definition of the reference standard incorporates results from the index tests.	The true infection status is defined based on both the RT-PCR and the antigen test result.
		Diagnostic review bias	The results of the reference standard are interpreted with knowledge of the results of the index test.	The results of the antigen test are known when the RT-PCR result is interpreted.
		Differential verification bias	Different reference standards are used between groups of patients to determine the true disease state.	Positive antigen test results are verified by RT-PCR and negative results are verified by a second antigen test.
		Partial verification bias	The true disease state is verified by a reference standard only for a subgroup.	Only positive antigen test results are verified by a RT-PCR.
Index test	The test of interest is called the index test, here being e.g. the antigen test.	Test review bias	The results of the index tests are interpreted with knowledge of the reference standard.	Antigen test results are interpreted with the knowledge of the RT-PCR result.
		Threshold bias	Diagnostic accuracy is evaluated for more than one threshold.	The threshold for the antigen test was not pre-specified.
Participant selection	Participant selection refers to the study population chosen for the respective study.	Selection bias	The study population is not representative of the population of the intended use.	Participants are selected based on results of RT-PCR.

		Spectrum bias	Not the whole spectrum of the disease is included.	Cases are patients with severe symptoms, and controls are drawn from pre-pandemic samples.
--	--	---------------	--	--

REVIEWERS' COMMENTS:

Reviewer #1 (Remarks to the Author):

The authors have addressed all my comments and I don't have further comments on the manuscript.

Reviewer #2 (Remarks to the Author):

The authors have addressed the comments, and it can be published as the current form.

We would like to thank the reviewers for the time and effort spent in reviewing our manuscript.

Madhav Chaturvedi, Denise Köster, Nicole Rübsamen, Antonia Zapf, and André Karch, on behalf of all authors.